# MEMBERSHIP LEAKAGE IN PRE-TRAINED LANGUAGE MODELS

## ABSTRACT

Pre-trained language models are becoming a dominating component in NLP domain and have achieved state-of-the-art in various downstream tasks. Recent research has shown that language models are vulnerable to privacy leakage of their training data, such as text extraction and membership leakage. However, existing works against NLP applications mainly focus on the privacy leakage of text generation and downstream classification, and the privacy leakage of pre-trained language models is largely unexplored. In this paper, we take the first step toward systematically auditing the privacy risks of pre-trained language models through the lens of membership leakage. In particular, we focus on membership leakage of pre-training data in the exposure of downstream models adapted from pre-trained language models. We conduct extensive experiments on a variety of pre-trained model architectures and different types of downstream tasks. Our empirical evaluations demonstrate that membership leakage of pre-trained language models exists even when only the downstream model output is exposed, thereby posing a more severe risk than previously thought. We further conduct sophisticated ablation studies to analyze the relationship between membership leakage of pre-trained models and the characteristic of downstream tasks, which can guide developers or researchers to be vigilant about the vulnerability of pre-trained language models. Lastly, we explore possible defenses against membership leakage of PLMs and propose two promising defenses based on empirical evaluations.

## 1 INTRODUCTION

Nowadays, pre-trained language models (PLMs), represented by BERT (Devlin et al., 2019), have revolutionized the natural language processing community (Wolf et al., 2019; Vaswani et al., 2017; Munikar et al., 2019). PLMs are typically pre-trained on large-scale corpora to learn some universal linguistic representations and are then fine-tuned for downstream domain-specific tasks (Sun et al., 2019; Shen et al., 2021). Concretely, downstream model owners can add only a few task-specific layers on top of the PTMs to adapt to their own tasks, such as text classification, named entity recognition (NER), and Q&A. This training paradigm not only can avoid training new models from scratch, but also form the basis of state-of-the-art results across NLP.

Despite its novel advantage in adapting downstream tasks, PLMs are essentially DNN models. Recent studies (Erkin et al., 2009; Liu et al., 2022; Choo et al., 2021; Li et al., 2021)have shown that machine learning models (e.g., image classifiers) are vulnerable to privacy attacks, such as attribute and membership inference attacks. Yet, existing privacy attacks against language models have mainly focused on text generation and downstream text classification (Song & Shmatikov, 2019; Shejwalkar et al., 2021). To our knowledge, the potential privacy risks of pre-training data for PLMs have never been explored.

To fill this gap, we take the first step towards systematically audit the privacy risks of PLMs through the lens of membership inference: An adversary aims to infer whether a data sample is part of PLMs' training data. In particular, given the realistic and common scenario that downstream service providers are more likely to build models adapted from PLMs, we consider that the adversary access only these downstream service models deployed online. Here, PLMs allow adding any task-specific layers to fit any type of downstream task, such as classification (Shejwalkar et al., 2021), NER (McCallum & Li, 2003), and Q&A (Bordes et al., 2014). We further consider another realistic scenario

where no additional information about the target PLMs is available to the adversary other than the output, i.e., black-box setting. We perform an extensive measurement study of membership inference, jointly, over four different PLMs architectures (BERT,ALBERT,RoBERTa, XLNet) and five different downstream datasets that refers to three downstream tasks. Our evaluations show membership leakage of pre-training data exists even when only the output of the downstream model is exposed, regardless of the PLMs architecture and downstream tasks,and thereby pose a more severe risk than previously thought. We further analyze the relationship between membership leakage and the characteristic of downstream tasks. We also conduct sophisticated ablation studies guideline inventors to be vigilant about the vulnerability of pretrained NLP models Lastly, we explore possible defenses that can prevent membership leakage of PLMs and propose two promising defenses based on empirical evaluation results.

**Contributions.**

- We pioneer to conduct the first investigation on membership leakage of PLMs' pre-training data with only downstream model output exposed.
- We conduct extensive experiments on a variety of PLMs architectures and different types of downstream tasks. Our empirical evaluations demonstrate that membership leakage of PLMs exists even when only the downstream model is exposed.
- We conduct sophisticated ablation studies to analyze the relationship between membership leakage and the characteristic of downstream tasks.
- We explore possible defenses against membership leakage of PLMs and propose two promising defenses.

## 2 BACKGROUND / RELATED WORK

### 2.1 PRE-TRAINED LANGUAGE MODELS

**Pre-trained Encoder.** Nowadays, large-scale pre-trained language models (PLMs) are pushing natural language processing to a new era. They are typically pre-trained on large corpora by self-supervised learning and fine-tuned for different types of downstream tasks. The first generation of PLMs was BERT Devlin et al. (2019), whose pre-training tasks were masked language modeling (MLM) and next sentence prediction (NSP). Since then, many variants of BERT have emerged to improve the learning ability of language models, such as ALBERT (Lan et al., 2020), RoBERTa (Liu et al., 2019), XLNet (Yang et al., 2019).

**Downstream Tasks.** PLM trained on a large corpus can learn generic linguistic representations, which can benefit in a wide range of downstream tasks such as text classification, named entity recognition (NER), and question and answering (Q&A). PLMs unify these tasks into a common pre-training and fine-tuning pipeline and achieves superior performance across them (Gururangan et al., 2020; Qiu et al., 2020), and thereby the pre-training and fine-tuning pipeline has become the most widely applied downstream model construction paradigm.

PLMs, despite their high level of generic linguistic representations, also rely on large-scale corpora that contains private/sensitive pre-training data, such as phone numbers, addresses, and biomedical data (in BioBERT (Lee et al., 2020)). Therefore, the vulnerability of PLMs to privacy leakage deserves our attention, as well as proactive assessment. Further, this is also important in light of the latest regulations under the EU General Data Protection Regulation (GDPR) umbrella[1] which require data owners to have greater control over their data. In addition, since PLMs serve downstream tasks, this means that downstream models adapted from PLMs are actually more common in the real world. Therefore, these concerns and realities drive our attention to the privacy leakage that exists in PLMs and that only the downstream models on them are exposed.

### 2.2 MEMBERSHIP INFERENCE ATTACKS

Membership inference attack is a kind of data inference attack which aims to infer whether the data sample was used to train the target machine learning model (Hu et al., 2021; Carlini et al., 2021;

---

[1] https://gdpr-info.eu

Ye et al., 2021). The core idea of membership inference is to leverage overfitting to conduct the attack. Usually machine learning models will overfit on its training data, so when querying the well-trained machine learning model with both member and non-member data, the model will have different behaviors, which could be leveraged to differentiate between member and non-members (Li & Zhang, 2021; He et al., 2022; Li et al., 2022; Carlini et al., 2021; Rahimian et al., 2020; He et al., 2022).

Existing attacks have been extensively studied in computer vision (Salem et al., 2019; He et al., 2022; Ye et al., 2021; Humphries et al., 2020; Liu et al., 2021; Yang et al., 2020), while only a few works have been done in NLP domain (Song & Shmatikov, 2019; Shejwalkar et al., 2021; Mireshghallah et al., 2022; Jagannatha et al., 2021). Especially in the NLP domain, existing work has focused on a specific downstream task, with Song & Shmatikov (2019) working in sequence models and Shejwalkar et al. (2021) working in text classification models. Mireshghallah et al. (2022); Jagannatha et al. (2021) work on masked language models but they can directly get access to target encoder's output, this assumption is not realistic. Further, all existing works can access models that are exactly trained or fine-tuned for the data they aim to infer without any inconsistency. In contrast, we consider a more realistic and challenging scenario where we do not have direct access to the original PLMs that is purely applicable to the pre-training data we aim to infer, i.e., there is actually an inconsistency between the PLMs now and the final output we can access. Second, this inconsistency is further amplified by the fact that the fine-tuning process will modify the parameters of the PLMs to fit the new downstream task. These inconsistencies make it more difficult to evaluate whether a model is truly vulnerable to membership inference or not, which may lead to premature claims of privacy for PLMs.

## 3 PROBLEM STATEMENT

### 3.1 PRE-TRAINED LANGUAGE MODELS

Here, we formulate a pre-trained language model. A pre-trained language model (PLM) that trained on large-scale pre-training dataset $\mathcal{D}$ can map textual inputs $t$ to embeddings. The downstream service provider then adds several layers on top of the PLM to build a new downstream model $\mathcal{M}$. The model $\mathcal{M}$ is then fine-tuned to fit their own downstream dataset by minimizing a predefined loss function using some optimization algorithms. Finally, the downstream model $\mathcal{M}$ can map textual inputs $t$ to new types of outputs, such as posteriors for text classification or embeddings for Q&A.

### 3.2 THREAT MODEL

**Attacker's Goal.** The goal of the attacker is to make a binary prediction of whether the given textual input is used to train the target pre-trained language model $\mathcal{M}_{target}$. A textual input is called a member (or non-member) of the $\mathcal{M}_{target}$ if it is in (or not in) the target pre-training data $\mathcal{D}_{target}$.

**Attacker's Background Knowledge.** Typically, the background knowledge in membership inference domain is along two dimensions: the architecture of target PLMs and the distribution of target pre-training dataset $\mathcal{D}_{target}$. In the first dimension, we assume that the attacker has no knowledge of the architecture of the target PLM and has only black-box access to the downstream model $\mathcal{M}_{target}$ adapted from the target PLM. The is the most realistic scenario as service providers typically share task-specific models adapted from PLMs to the public as "machine learning as a service", leading the attackers to access the output of the downstream model $\mathcal{M}_{target}$ directly rather than the output of the target PLMs to launch membership inferences.

Besides, we assume that the attackers has a very small subset from the member training data of pre-training data $\mathcal{D}_{target}$, as well as a small set of local non-member data $\mathcal{D}_{local}^{non\_member}$. The adversary then constructs an auxiliary dataset $\mathcal{D}_{auxiliary} = \{x^m \cup x^{nm} : x^m \in \mathcal{D}_{sub\_target}^{member}, x^{nm} \in \mathcal{D}_{local}^{non\_member}\}$ that can be used to train the attack model $\mathcal{A}$. Note that, the assumption of auxiliary dataset also holds for previous works (Liu et al., 2022). In addition, since the PLMs are usually trained on billions of pre-training data, it is very likely that an attacker could collect a very small fraction of such the large-scale pre-training data, which is also more reasonable and efficient than collecting another billions of local corpora to build a local shadow model. Last but not least, attacks on real-world models are also more realistic than existing works in laboratory settings.

## 4 METHODOLOGY

### 4.1 INTUITION

The key intuition of our work is a general observation of PLMs' memorization on pre-training data. Given that PLMs can map textual inputs $t$ to embedding vectors for the benefit of downstream tasks, PLMs are typically trained deeply on pretrained data $\mathcal{D}_{target}$ to learn general language representations, i.e., PLMs memorize pre-training data deeply. By further analogy with the fact that the confidence scores of members in image classification are usually larger than non-members[][][], this yields the following key intuition about membership information:

- Members and non-members of PLMs behave differently on embeddings.
- Due to deep memorization, such behavioral differences on embedding vectors will be retained in the downstream model output.

Thus, we can use the retained differences reflected in the downstream model output (although more likely not as much as the embedding vectors directly output by the PLMs) to distinguish between members and non-members of the PLMs.

### 4.2 ATTACK METHODOLOGY

Here, we propose the methodology to evaluate the membership leakage of PLMs.

**Auxiliary Dataset Construction.** As aforementioned, we assume that an attacker can collect a very small subset contained in the pre-training data $\mathcal{D}_{target}$, i.e., the members of PLMs. The attacker then constructs an auxiliary dataset by combining all these collected datasets

**Attack Training Dataset Construction.** The attacker feeds all samples of the auxiliary dataset to the downstream model and records all outputs. The attacker then constructs the training dataset for the attack model by labeling these recorded outputs as members or non-members, respectively.

**Attack Model Training.** The attacker build an attack model, i.e,. a binary classifier, and then train it with the auxiliary dataset in conjunction with classical training techniques.

**Membership Inference.** Finally, the attacker can feed a candidate text input into a well-trained attack model, and then the attacker determines that the candidate text input is a member or non-member based on the predictions of the attack model.

## 5 EVALUATIONS

### 5.1 EXPERIMENTAL SETUP

**PLM Architectures and Pre-training Datasets.** We consider the four recent state-of-the-art PLMs: BERT (Devlin et al., 2019), ALBERT (Lan et al., 2020), RoBERTa (Liu et al., 2019), XL-Net (Yang et al., 2019). These PLMs are constructed from different loss functions and training schemes as well as architectures. Note that due to the resource limitation, i.e., insufficient GPU resource, we are unable to train our PLMs on billions of corpora. Thus, we use well-trained PLMs publicly available online as our attack targets[2]. Actually, this is more realistic than almost all the existing works only in the laboratory environment.

Each PLM application benchmarks its own huge amount of pre-training data. Here, we emphasize that two datasets, i.e., Wikipedia and BooksCorpus (Devlin et al., 2019), are the most popular and are used across all these four PLMs, and consider them as members in our work.

**Downstream Models and Fine-tuning Datasets.** As mentioned earlier, we considered the most realistic and challenging scenario where we can only access the output of downstream models adapted from PLMs and can design downstream models for any type of downstream task. Without losing generality, we considered six benchmark datasets to cover three trendy topics, namely SST/AG's News/Yelp Review Full for text classification (Socher et al., 2013; Zhang et al., 2015), CoNLL2003

---

[2]All four models we consider in this paper are downloaded from https://huggingface.co/models

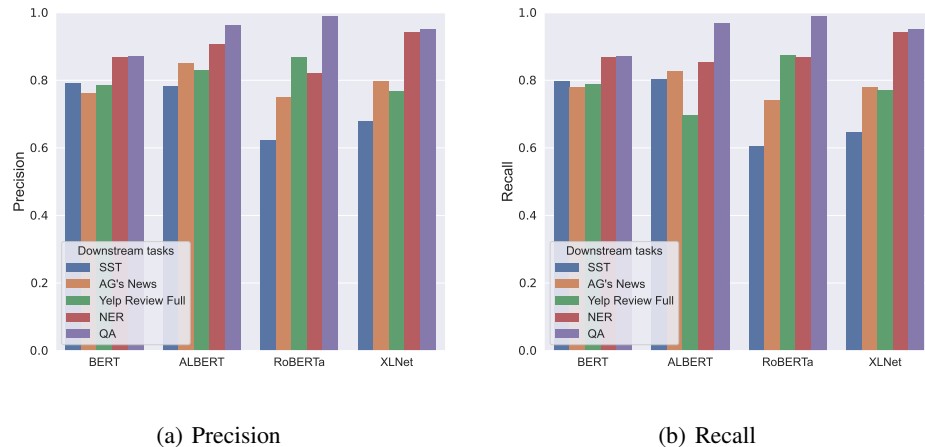

(a) Precision             (b) Recall

Figure 1: Attack performance on text classification, NER and Q&A tasks. SST/AG's News/Yelp Review Full are three text classification tasks with 2/4/5 classes; CoNLL2003 is NER dataset; SQuAD v1.0. is Q&A dataset.

for Named Entity Recognition (NER) (Li et al., 2020), and SQuADv.10 & one for Q&A (Bordes et al., 2014).

**Auxiliary Dataset and Attack Model.**   To build the auxiliary dataset, we select a very small subset from the pre-training data used for all these PLMs, i.e., 0.23% of Wikipedia and 0.02% of BooksCorpus, as members. We then select third-party datasets as non-member datasets, which is different from both the pre-training and fine-tuning datasets. The non-member datasets are IMDB, CoLA and AX, which are from the GLUE benchmark dataset (Devlin et al., 2019). We use a mix of the three datasets as non-members for better generalization.

For attack model, we build a three-layer MLP as the attack model, which accepts the output of downstream models as input. Due the various of downsteram task, the dimentions of fist layer of attacks is also different across different tasks.

**Evaluation Metrics.**   We use precision and recall as evaluation metrics for attack performance, as used in previous works (Salem et al., 2019; He et al., 2022).

**Attack Implementation.**   We train the attack model for 100 epochs with a learning rate of 1e-5, and the optimizer we use is the Adam (Kingma & Ba, 2015) optimizer. We split the auxiliary dataset into training and testing data for the attack model in a 5:1 proportion. We also further investigate the effect of the size or type of the auxiliary dataset on the attack performance in Section Section 5.2.

## 5.2 EXPERIMENTAL RESULTS

**Attack Performance.**   First, we report the attack performance, i.e., precision and recall, in Figure 1 We can observe that membership inference achieves significant performance across all PLMs and different downstream tasks compared to random guessing. (i.e., 0.5). For instance, the precision of BERT fine-tuned on SST/NER/Q&A is 0.774/0.875/0.873, and the recall is 0.825/0.862/0.870, which is much higher than the random guess. These results indicate that membership leakage in PLMs exists regardless of the type of downstream task, which poses a much more serious threat to PLMs than that shown by existing work. These empirical results verify our key intuition quantitatively.

**Membership Leakage Analysis.**   The above results fully demonstrate the vulnerability of PLMs to membership leakage. Here, we delve more deeply into the reason behind that from a visualization method. We study the pre-trained BERT that is then fine-tuned on AG's News dataset. We randomly sample 1000 member samples and 1000 non-member samples. We first feed these samples to the original pre-trained BERT and embed the output into 2D space using t-Distributed Stochastic

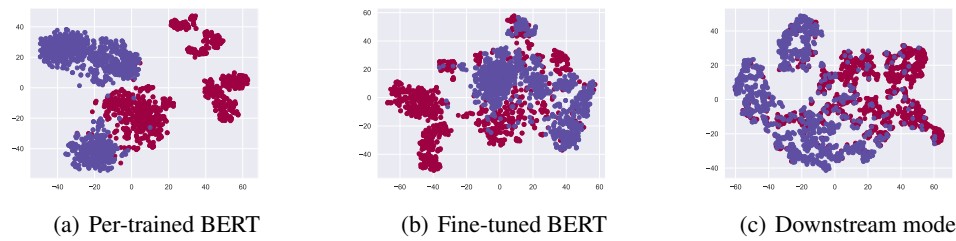

(a) Per-trained BERT      (b) Fine-tuned BERT      (c) Downstream model

Figure 2: TSNE visualization on BERT fine-tuned on AG's News. Each plot means that TSNE visualization is based on the embedding of this model.

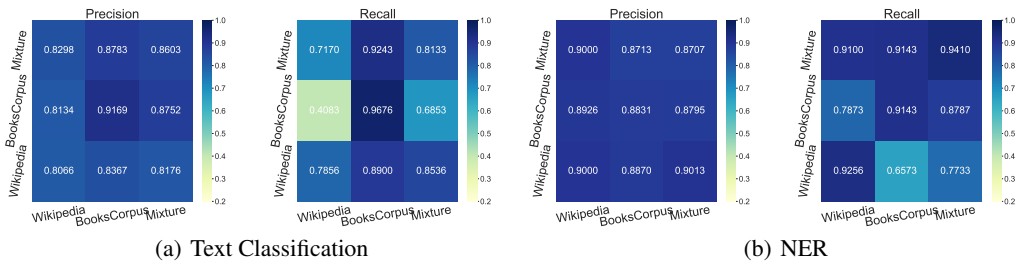

(a) Text Classification                  (b) NER

Figure 3: Attack performance with relaxation of member datasets, the x-axis represents the datasets used for training attack model, the y-axis represents the datasets used for testing attack model

Neighbor Embedding (t-SNE)[3], as shown in Figure 2(a) Similarly, for the downstream model, i.e., by adding a few layers on top of the pre-trained BERT and fine-tuning on AG's News, we obtain t-SNE visualizations of the BERT and downstream model outputs, as shown in Figure 2(b) and Figure 2(c), respectively. From these results, we can make the following observations:

- Members and non-members on original PLMs behave differently.
- Members and non-members on fine-tuned PLMs still behave differently.
- Encouragingly, members and non-members on the fine-tuned downstream model also behave differently, though not as much as on PLMs.

These observations qualitatively reveal that membership leakage of PLMs' pre-trained data persists, although only downstream models are exposed and PLMs are not, and again verify our key intuition.

**Relaxation of Assumptions on Auxiliary Dataset.** As aforementioned in Section 5.1, we use a mixture of Wikipedia and BooksCorpus datasets as our members, a mixture of three third-party datasets as our non-members, and then construct the auxiliary dataset. Here, we relax the assumption of members and non-members the adversary can access:

- `Relaxation-I`: The adversary obtains only one of the members, e.g., Wikipedia, and uses it to train the attack model and finally evaluate the attack performance on the other unseen members i.e., BooksCorpus.
- `Relaxation-II`: The adversary obtains only one of the non-members, e.g., IMDB, and uses it to train the attack model and finally evaluate the attack performance on the other unseen non-members, i.e., AX or CoLA.

Note that when we focus on the relaxation of one of the assumptions, we keep the other assumptions unchanged as described in Section 5.1.

We report the attack performance for `Relaxation-I` in Figure 3. First, we can observe that the attack performance remains very high for all different types of downstream tasks. These results

---

[3]https://github.com/DmitryUlyanov/Multicore-TSNE

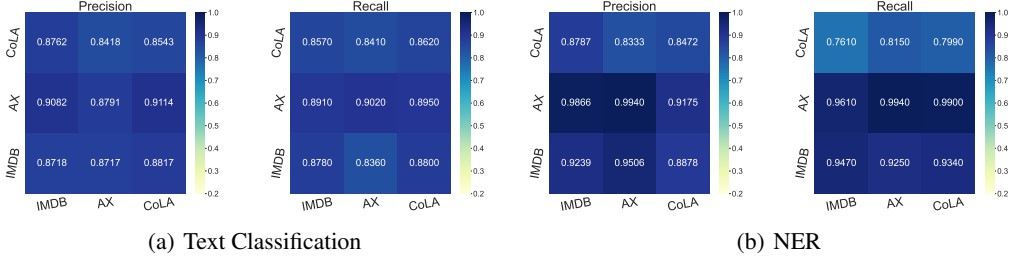

(a) Text Classification        (b) NER

Figure 4: Attack performance with relaxation of non-member datasets, the x-axis represents the datasets used for training attack model, the y-axis represents the datasets used for testing attack model

show that whatever the type of members, they behave differently from non-members, and that such differential behavior between different types of members and non-members share the same or similar boundary. In other words, even if an adversary collects only one certain types of members, he/she can still obtain significant attack performance, which extends the scope of member inference attack.

We report the attack performance for `Relaxation-II` in Figure 4. We can observe that the attack performance is still very high for all different types of downstream tasks. Similarly, these results show that whatever the type of non-members, they behave differently from members, and that such differential behavior between different types of non-members and members share the same or similar boundary. Even if an adversary collects only one certain types of non-members, he/she can still obtain significant attack performance, which further extends the scope of member inference.

In summary, these empirical results demonstrate that membership leakage of PLMs is much more severe than previously thought.

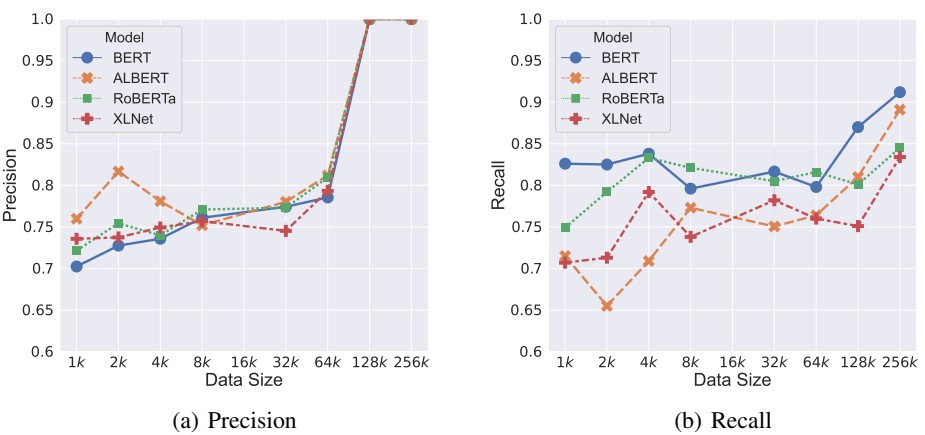

(a) Precision        (b) Recall

Figure 5: Attack performance versus the size of member dataset

**Effect of Member Data size.** We further investigate the effects of size of member data the adversary collect. In our default setting, we adopt 0.23% of Wikipedia and 0.02% of BooksCorpus as members, i.e., 1.5w and 1.5w, respectively. Here, we evaluate the attack performance with different size of members by varying it from 1k to 256k. We report the attack performance across different PLMs fine-tuned on AG's News. As we can see in Figure 5, there is a clear trend of increasing precision and recall as the member size increases. These results actually are consistent with the expectation that more training data leads to high performance for an ML model, whereas the attack model is essentially an ML model. Encouragingly, we can observe that even though the size of members is only 1k, that is, 500 from Wikipedia and 500 from BooksCorpus, the attack performance is still very high. Note that 500 samples only account for 0.0077% of Wikipedia and 0.00067% of

BooksCorpus. Such a low proportion gives developers or researchers a warning that membership inference against PLMs is easier to start, which once again poses a serious threat. Besides, they should carefully release their pre-training data to prevent their training information from leaking.

**Effect of Number of Classes.** Then we study the relationship between membership inference and the number of classes of downstream tasks. The data set we use here is Yelp Review Full Dataset (Zhang et al., 2015), which is an online review data set for business and has five classes. Here, we construct 2-class dataset, 3-class dataset, 4-class dataset and 5-class dataset extracted from Yelp Review Full dataset. The only difference between these extracted datasets is the number of classes, and the size of these datasets is the same. As shown in Figure 6, the attack performance increases with the number of classes, which indicates that the more classes, the more membership information leaked by the downstream model. A straightforward reason is that the dimension of the downstream model's output increases with the number of classes, which brings more information about membership, thus leading to higher performance of membership inference.

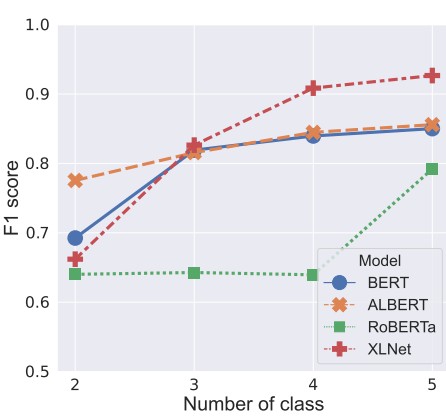

Figure 6: Attack performance versus number of classes.

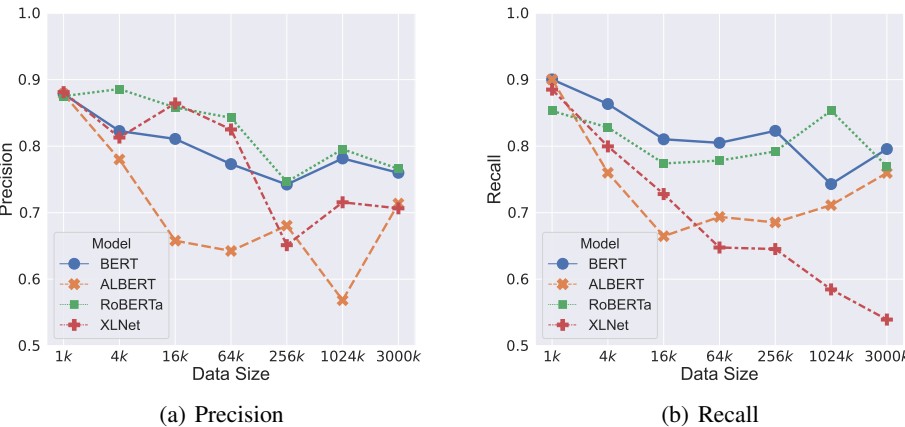

| (a) Precision | (b) Recall |
|---|---|

Figure 7: Attack performance versus different fine-tuning data size

**Effect of the Size of Fine-tuning Dataset.** Here, we study the effect of the size of fine-tuning dataset. We random sample different sizes of samples from Amazon dataset to fine-tune the all the PLMs we consider in this paper. The attack performance in Figure 7 shows that attack performance drops with the increase of fine-tuning data size. The reason is that adding more fine-tuning data to fine-tune the model is equal to adding additional knowledge to the PLMs, and PLMs will change to a greater extent, so the membership information may be laid over by the fine-tuning process, which leads to worse attack performance.

**Effect of Number of Epochs.** Lastly, we study the effect of the number of fine-tuning epochs. As shown in Figure 8, attack performance decreases as the number of epoch increases. PLMs change to a larger extent as the number of fine-tuning epochs increases, so membership information of pre-training data is more likely to be forgotten, leading to poorer attack performance. However, in practice, download service providers typically fine-tune their task-specific models within no more

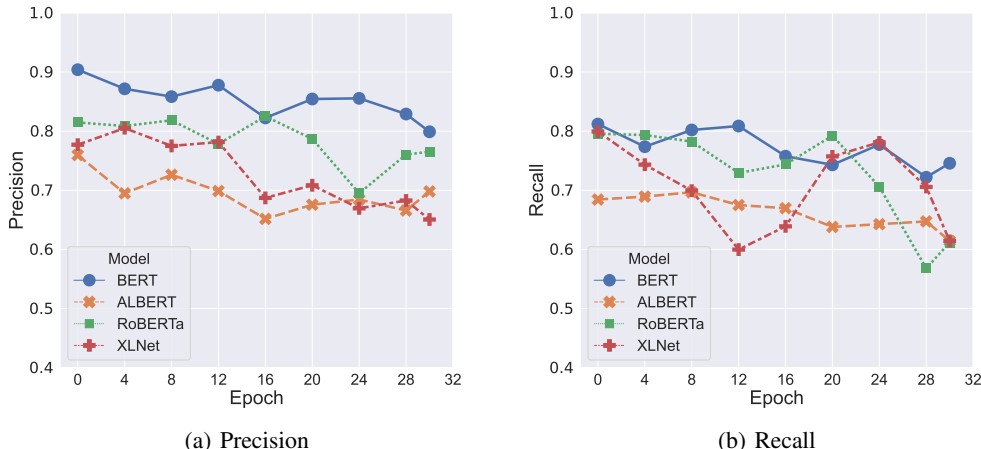

(a) Precision            (b) Recall

Figure 8: Attack performance versus different number of fine-tuning epochs

than 10 epochs, which means that membership leakage of PLMs is resisted after the fine-tuning process.

## 6 POSSIBLE DEFENSE

To mitigate the threat of membership leakage, a large body of defense mechanisms have been proposed in the computer vision domain, such as data augmentation, Differential privacy (Acs et al., 2017; Chen et al., 2020; Bun & Steinke, 2016), adversary regularization (Nasr et al., 2018), Mem-Guard (Jia et al., 2019). Note that due to the resource limitation, i.e., insufficient GPU resource, we are unable to train our PLMs on billions of corpora, so we adopt the pre-trained language models publicly available online. Therefore, we cannot evaluate the first three defenses that need to be applied in the pre-training process. For MemGuard that changes the confidence score of a classification task, it definitely achieves good defense performance in downstream classification tasks. However, for NER and Q&A, the output vectors contain enough information for their own downstream tasks, and MemGuard that changes the output vectors can lead to performance degradation.

In addition to existing defenses, we here explore two possible defenses based on the above evaluation in Section 5.2, i.e., increasing the size of the fine-tuning dataset and increasing fine-tuning epochs. As shown in Figure 7 and Figure 8, we can observe that the attack performance decreases with the increase of the size of fine-tuning dataset and fine-tuning epochs. Both can change PLMs to a greater extent, so they can override the membership information through fine-tuning process, which leads to the reduction of membership leakage. In addition, both possible defenses also benefit the downstream task, i.e., improving the utility performance of the downstream model. Lastly, we leave it as future work to explore effective defenses against membership leakages of PLMs.

## 7 CONCLUSION

In this work, we take the first step to systematically evaluate the membership leakage of pre-training data of PLMs. In particular, we consider a realistic scenario where the adversary can only get access to the output of downstream models adapted from PLMs. We conduct extensive experiments on a variety of PLMs architectures and different types of downstream tasks. The experiment results shows membership leakage of PLMs exists even when only the downstream model is exposed, showing a severe threat than previously thought. To further study which factors that affect membership leakage, we conduct an in-depth ablation study from different angles, which can direct model inventors to be alert to vulnerabilities of PLMs. Finally, we explore possible defenses against membership leakage of PLMs and propose two promising defenses.

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
