# OpenReview forum: "Membership Leakage in Pre-trained Language Models"
_ICLR.cc/2023/Conference — Submitted to ICLR 2023_

### Official Review · Reviewer_h5pk · 2022-10-16

**Confidence:** 3
**Correctness:** 1
**Technical Novelty And Significance:** 2
**Empirical Novelty And Significance:** 1
**Recommendation:** 3

**Clarity, Quality, Novelty And Reproducibility:**

Some typos I found: 1) The sixth row of Introduction, ‘PTM->PLM’; 2) One period is missing in Line 9 of Page 2; 3) Several citations are missing in Section 4.1.

**Strength And Weaknesses:**

Evaluating the privacy risk of pre-training data through fine-tuned models is an interesting problem. I think in reality it is possible that the pre-trained language models are held as secret and the attacker  could one access fine-tuned APIs. Exploring this direction would help the community to understand how the privacy risk of pre-training data changes after fine-tuning.

However, the ‘membership inference attack’ in this paper is not a qualified attack because of the choice of attack model’s training data. The member/non-member data used to train attack model have to be from **the same distribution**. The non-membership data in this paper are data from the GLUE benchmark which are not from the same distribution as the pre-training data. The ‘attack’ then becomes a standard text classification task. A proper non-member data should be from the same source as the pre-training data, e.g., the test set of the pre-trained model.


Here are some other suggestions.

1. “To our knowledge, the potential privacy risks of pre-training data for PLMs have never been explored.” There is a line of works focus on the privacy risk of pre-trained models. A representative one is Carlini et al., 2021[1]. It directly attacks pre-trained models instead of fine-tuned ones. Therefore, this claim is inaccurate.

2. How do you set the number of member datapoints to train the attack model? You choose $0.23\%$ of Wikipedia and $0.02\%$ of BookCorpus. Why these two specific numbers


[1]: Extracting Training Data from Large Language Models, https://arxiv.org/abs/2012.07805.


**Summary Of The Paper:**

This paper runs inference attacks against fine-tuned language models to evaluate membership leakage of pre-training data. The attacker is assumed to have access to a small subset of the pre-training dataset. The attacker trains an attack model whose training data is the subset of pre-training data and some non-member data.

**Summary Of The Review:**

My recommendation is mainly because the attack in this paper is not a qualified membership inference attack. It is more like a text classification task because of the choice of training data for the attack model.

---

### Official Review · Reviewer_EhJ8 · 2022-10-22

**Confidence:** 5
**Correctness:** 1
**Technical Novelty And Significance:** 1
**Empirical Novelty And Significance:** 1
**Recommendation:** 1

**Clarity, Quality, Novelty And Reproducibility:**

I consider the novel of this paper moderate. There was membership inference or data extraction attacks against language models (e.g., "Extracting Training Data from Large Language Models" by Carlini et al). Also, there is a large body of membership inference attacks in other domains. However, this paper considers a fine-tuned model to infer the pre-training sentence membership, which is a slightly different setting.

The writing can be improved a lot. Here are some examples.
- Typos: PTM, guideline, missing citations rendered [][][], fist.
- 'Local' and 'sub_target' are not defined in the definition of non member dataset.
- Fig 2: The colors are not explained.
- The meaning of "1.5w and 1.5w" is not clear.

Additional comments.
- Why using Wikipedia as train is worse than Bookcorpus when the test data is Bookcorpus in Fig 3? Similarly, AX is the best for all test datasets.
-How about more traditional differential privacy to the output embedding?

**Strength And Weaknesses:**

Strengths.
- S1. The proposed classifier achieves better precision and recall than random guess.
- S2. The experiment considers dimensions such as the number of epochs and the size of the fine-tuning dataset.
- S3. A defense is discussed.

Weaknesses.
- W1. The claim of higher certainty by the fine-tuned model with a member sample is not verified. Instead, a black box classifier is trained. This hinders readers from understanding how this is made possible.
- W2. The attack model might be classifying the sentence type (movie review, Q&A, ...) rather than the membership, as they are consistently coming from certain sources, and they have distinct styles, semantics, and category. So the experiment is not well controlled to say the cause is the membership. For example, the positive dataset includes Wikipedia which is very general and encyclopedic, but IMDB is a review dataset. The fine-tuned model with SST will classify the IMDB data into positive and negative, while Wikipedia sentences often do not have clear polarity. Similarly, the output of the fine-tuned model with AG's News can provide the category of the input sentence, and this setting of Wiki/Book vs. Movie review, etc dataset can be easily classified with such category information. Again, knowledge vs. review data can have very different semantics and styles. Thus, the cause of the precision and recall is unclear. For more accurate assessment, a pre-training has to be done on the same type of corpora, or even better, by dividing a single corpus into two at the pre-train stage to properly test them.
- W3. The absolute number for the fraction of the known members 0.23% and 0.02% is not given initially, which is seemingly very large considering the size of Wikipedia and BookCorpus.


**Summary Of The Paper:**

This paper introduces a membership inference attack against a fine-tuned language model where the targets are the sentences from the pre-train corpus. The attack trains a simple classifier with known members and non-members using the outputs of the fine-tuned model. The empirical evaluation using Wikipedia and BookCorpus as a positive set and SST, AG News, and Yelp Review as a negative set. The classifier could identify these two sets with higher precision and recall than random guess.

**Summary Of The Review:**

This paper considers an interesting problem. Membership inference in other domains show the task is actually very difficult. In contrast, the experimental result in this paper looks promising despite the additional fine-tuning step. However, I doubt the cause is properly understood and the variables are correctly controlled. The paper unfortunately states that pre-training cannot be controlled due to the resource, but proper evaluation can be only done when the pre-train corpus is at least split into a train dataset for pre-train, a train dataset for the membership classifier, and a membership test dataset, to control the style, category and overall semantic of the test datasets. Without these, the membership classifier can look at which dataset the test sentence came from using the output of the fine-tuning tasks, which conveniently related to the chosen negative dataset. There was no attempt to understand or compare the outputs of the fine-tuned model either, making the verification of the claim not possible.

---

### Official Review · Reviewer_qzMP · 2022-10-24

**Confidence:** 4
**Correctness:** 2
**Technical Novelty And Significance:** 2
**Empirical Novelty And Significance:** 3
**Recommendation:** 5

**Clarity, Quality, Novelty And Reproducibility:**

I have extensively expressed my comments above. However, to summarize, I believe the paper needs some re-writing to better position it and improve its clarity.

In terms of novelty, I think the paper is novel in terms of what it studies, but not in terms of the methods used.


The paper seems fairly reproducible to me, if writing is clarified.

**Strength And Weaknesses:**

Strengths:
1. The paper studies downstream fine-tuning tasks, and measures leakage of pre-training data on those which I haven't really seen studied before. I have seen papers measuring memorization of fine-tuning data after pre-training, and papers studying pre-training alone, but not this scenario which is interesting.

2. I like how the authors study and ablate the effect of different things, such as training epochs, classes, dataset size, etc.


Weaknesses:
1. The paper needs to clarify/narrow its claims and assumptions a bit better. More specifically, the following issues need to be clarified:

a.	Use of the term pre-trained LMs, or PLMs: The opening paragraph of the paper states "... (PLMs), represented by BERT" which is not accurate, as BERT only represents MLMs as pre-trained models, whereas there is a whole different class of generative (auto-regressive), pre-trained models, such as the GPT family, whose memorization and leakage has been widely studied before [1-3]. I think it is really important that  the paper clarifies what it means by pre-training early on and not use the term pre-trained LM solely to BERT-based models (also encoder-decoder pre-trained models such as T5 and BART are also fairly common and are referred to as pre-trained LMs).

b.	Claiming prior work has not studied privacy risks/memorization in pre-training: I think this paper is very interesting and the problem is novel, however the positioning right now is inaccurate and needs to be improved. the claim "Pre-training has never been studied" needs to be better clarified: In terms of memorization of pre-training data,  Carlini et al. [1] study it for the GPT models, [4-5] study it for MLMs, and [2] studies memorization in LLMs. There is also work studying the memorization of fine-tuning LMs [3,6]. What the paper is studying is leakage of pre-training data through downstream tasks. This needs to be better clarified.

c.	In the sentence "PLMs memorize pre-training data deeply. By further analogy with the fact that the confidence scores of members in image classification are usually", instead of image classification, NLP papers can be used as evidence as there are already many NLP attack papers.


2. The writing and representation of the results is a bit confusing. For example, it is really hard for me to figure out what exactly is the downstream task in Figure 2? Another issue is what results are reported over the downstream task data and what membership results is over pre-training data? For example, the number of classes experiment at the end of the paper, is it the effect of number of downstream classes on pre-training data memorization/forgetting? It seems not. However, the number of training epoch experiments seems to study pre-training data forgetting as fine-tuning goes on. If the paper focuses on pre-training data, it should make that clear and present results accordingly.

3. I also don't understand how Figures 3 and 4 show "In summary, these empirical results demonstrate that membership leakage of PLMs is much more severe than previously thought.", as in what was previously thought? The fact that fine-tuning/downstream classification memorizes samples a lot was shown before so I am not exactly sure what this conclusion is.

4. In Figure 2's explanation, the paper says: "Encouragingly, members and non-members on the fine-tuned downstream model also be- have differently, though not as much as on PLMs.", however, the points actually overlap a lot. I would actually be interested in seeing the precision-recall numbers for this experiment, as I think this is the most important experiment in the paper, which directly studies leakage of pre-training data after fine-tuning.


Minor:
1.	”machine learning as a service”, the opening quotation marks need to be fixed
2.	"that the attackers has a very small subset" -> have
3.	"Page 4: larger than non-members[][][]," references are missing

References:
[1]  Carlini, Nicholas, et al. "Extracting training data from large language models." 30th USENIX Security Symposium (USENIX Security 21). 2021.

[2] Tirumala, Kushal, et al. "Memorization Without Overfitting: Analyzing the Training Dynamics of Large Language Models." arXiv preprint arXiv:2205.10770 (2022).

[3] Mireshghallah, Fatemehsadat, et al. "Memorization in NLP Fine-tuning Methods." arXiv preprint arXiv:2205.12506 (2022).

[4] Mireshghallah, Fatemehsadat, et al. "Quantifying privacy risks of masked language models using membership inference attacks." arXiv preprint arXiv:2203.03929 (2022).

[5] Lehman, Eric, et al. "Does BERT Pretrained on Clinical Notes Reveal Sensitive Data?." arXiv preprint arXiv:2104.07762 (2021).

[6] Shejwalkar, Virat, et al. "Membership inference attacks against nlp classification models." NeurIPS 2021 Workshop Privacy in Machine Learning. 2021.




**Summary Of The Paper:**

This paper focuses on the leakage of pre-training data, after fine-tuning for down-stream tasks, through the downstream task API. In other words, this paper studies how much pre-training data a fine-tuned model leaks, if black-box access is provided to a fine-tuned pre-trained model. The paper assumes access to 'auxiliary' data, which is a combination of a few samples of pre-training data, and non-member data, which are all fed to the down-stream model, logits collected and then used as 'training data' for an attack model, which has the binary task of guessing membership based on downstream model's logits.  The paper then mounts this attack on different scenarios with different setups and studies memorization and shows that pre-training data does indeed leak from the downstream models.

**Summary Of The Review:**

I have provided thorough feedback in the strengths/weaknesses box which includes my questions. I am willing to increase my score if my questions are answered, specifically the confusion about where membership is measured over fine-tuning data and where it is measured over pre-training data.

---

> ### Author Response · Authors · 2022-11-16
> **Response to reviewer qzMP**
>
> We are really grateful for the reviewer's expertise for pointing some concerns of the paper in detail. This will help improve this work a lot.
>
> For the first issue about the term of PLMs, thanks for your expertise and full knowledge about the nowadays pre-training framework in NLP. I will improve this part according to your advice and rewrite the paper. I will clarify in this paper that our pre-training is based on masked language models to distinguish it from other pre-trained language models.
>
> For the second issue about related works, thanks again for the reviewer's professional knowledge about the membership leakage in NLP pre-training models and the precise control of our work's core idea. I will clarify this work is studying the membership leakage of pre-training data through the downstream tasks as you mentioned.
>
> For the third issue, sorry to make you confused. The paper focuses on pre-training data no matter what downstream tasks factor it studies. In Figure 2 the downstream task is a 4-class text classification task which is fine-tuned on AG's News.  In the number of classes experiment, yes we indeed studied the effect of the number of downstream classes effect on the pre-training data's memorization/forgetting. The target data we aim to infer or evaluate is always the pre-training data and we want to know different downstream factors' effects on pre-training data's memorization/forgetting. Thanks for your suggestions and I will improve this part to make it more clear.
>
> Regarding the concern about Figure 3 and Figure 4, sorry to make you confused again. Here the attack model in Figure3/4 only uses one kind of member(non-member) dataset for training the attack model and testing its performance on other unseen members (non-members), the membership inference attack's success in this scenario shows that the data used for training and testing attack model may be different datasets and thus have different distributions. This is different from the previous work where the data used for the attack model's training and testing are from the same dataset. This also extends the attack scenario. Considering this, we claim that the threat is more severe than before.
>
> I am really grateful for your time and efforts to propose these valuable suggestions and comments. I will polish my paper according to your professional review.

---

### Decision · Program_Chairs · 2023-01-20

**Decision:**

Reject

**Justification For Why Not Higher Score:**

Generally the reviewers have recommended a reject quite overwhelmingly.

**Justification For Why Not Lower Score:**

N/A

**Metareview: Summary, Strengths And Weaknesses:**

The paper presents some methods towards detecting membership leakage of pre-training data in the exposure of downstream models adapted from pre-trained language models.  The work focuses on four different pre-trained models: BERT, ALBERT, RoBERTa and XLNet and several downstream tasks.  The experiments show that membership leakage of pretraining data happens even while only using models tuned on downstream tasks.

Strengths: The reviewers and I agree that the paper studies the problem of pretraining data leakage after fine tuning on downstream models, which is novel.

Weaknesses:  The reviews have more details, but there is consensus that the paper overclaims and does not appropriately situates this work in comparison to prior work, also provides experimental results on only one family of MLM models, and has issues with presentation.